

# BinBench: a benchmark for x64 portable operating system interface binary function representations

Francesca Console, Giuseppe D'Aquanno, Giuseppe Antonio Di Luna and Leonardo Querzoni

Department of Computer, Control and Management Engineering, University of Roma "La Sapienza", Rome, Italy

## ABSTRACT

In this article we propose the first multi-task benchmark for evaluating the performances of machine learning models that work on low level assembly functions. While the use of multi-task benchmark is a standard in the natural language processing (NLP) field, such practice is unknown in the field of assembly language processing. However, in the latest years there has been a strong push in the use of deep neural networks architectures borrowed from NLP to solve problems on assembly code. A first advantage of having a standard benchmark is the one of making different works comparable without effort of reproducing third part solutions. The second advantage is the one of being able to test the generality of a machine learning model on several tasks. For these reasons, we propose BinBench, a benchmark for binary function models. The benchmark includes various binary analysis tasks, as well as a dataset of binary functions on which tasks should be solved. The dataset is publicly available and it has been evaluated using baseline models.

# BACKGROUND

## Introduction

Deep neural networks (DNNs) are the tool of choice for solving problems in several fields, to name a few: natural language processing, image processing, audio classification. This is due to their ability to solve complex problems using a purely data driven approach. Following this trend the research community of binary code analysis is successfully applying DNNs to the solution of several challenging tasks that range from binary similarity to the reconstruction of stripped symbols. As hinted by the naturalness hypothesis of *Allamanis et al. (2018)*, the code, also in its binary form, is a mean of communication between humans and machine. An empirical confirmation of this hypothesis is the fact that many articles using DNNs on the binary domain are using architectures that are borrowed, often with minimal modifications, from the NLP community (examples are the self-attentive RNN used in *Massarelli et al. (2019b)*, word2vec used in *Chua et al. (2017)*), the transformer based architecture used in *Li, Yu & Yin (2021)*. Unsurprisingly, also in this field, DNNs are showing state of the art performance.

Corresponding author
Francesca Console,
fconsole@diag.uniroma1.it

A defining aspect of DNNs is that they learn an inner representation of the objects they are analysing, that is then used to solve the task at hand. In this regard, the NLP community has built sophisticated architectures, such as BERT (*Devlin et al., 2019*), GPT-2 (*Hegde & Patil, 2020*) and GPT-3 (*Brown et al., 2020*), these DNNs are able to learn complex representations of the sentences they are analysing and are able to use these to solve a panoply of tasks, acting as *universal unsupervised task solvers* (*Young et al., 2018*; *Radford et al., 2019*). For their generality, these architectures have to be tested on several different tasks. A key achievement in the NLP community has been the adoption of a set of standard datasets (*e.g.*, SQUAD (*Rajpurkar et al., 2016*), GLUE (*Wang et al., 2018*), GLUECoS (*Khanuja et al., 2020*), and KILT (*Petroni et al., 2021*)) that are used to evaluate the performance of new architectures on several different tasks. As an example, the GLUE benchmark contains various tasks associated to different datasets, such as *Corpus* of Linguistic Acceptability (CoLA), Stanford Sentiment Treebank (SST-2) and Quora Question Pairs (QQP). In detail, CoLA asks to understand whether samples are grammatical English sentences. The task of SST-2, instead, consists in predicting the sentiment of each analyzed sentence. Finally, QQP asks to understand whether two given questions are semantically equivalent. These multi-task benchmarks are designed to evaluate the generality of any newly proposed architecture.

The binary analysis community working with and on DNNs is still lacking the adoption of a common multi-task benchmark that could be used to test deep architectures on binary analysis tasks. As we will discuss, the availibility of a widely-recognized benchmark for DNN-based binary analysis is a crucial aspect that has been mainly neglected so far. All the articles present in the literature are using their own datasets (see *Massarelli et al., 2019b*; *Xu et al., 2017*; *Ding, Fung & Charland, 2019*; *Artuso et al., 2021*; *Massarelli et al., 2019a*), specifically created to solve the single problem at hand.

This raises two concerns. The first issue is that results between works are not comparable. Performances could, and often do, change drastically when using different datasets. This makes hard or impossible to decide which architecture to pick for a certain problem. The only way to get comparable results is to run different candidate solutions on the same dataset. Unfortunately, this is far from trivial: when datasets and prototype solutions are not publicly released, it requires a consistent effort to replicate data needed for a fair comparison.

The second issue is more subtle, state of the art solutions in NLP are general neural networks pipeline (*e.g.*, the transformers) that during training learn a general representation of natural language. Such architectures, usually after a fine tuning step, can be used to solve several different tasks. The current trend in the binary analysis community is to design an *ad-hoc* DNN architecture for each single problem. However, a worthwhile effort is the of creating an efficient DNN architecture that can learn and use a representation of a binary function, and then use this representation to solve several different tasks.

To test such a system, a multi-task benchmark is needed, to assess the performance of a candidate architecture trained to learn a binary code representation on several tasks focused on different facets of the binary analysis problem: from syntax-related tasks (*e.g.*,

understanding which compiler has generated a binary code snippet), to semantic-related ones (*e.g.*, identifying what a function is doing).

In this article we propose a multi-task benchmark for testing the generality of DNN architectures. We hope this work will be the first step towards the creation of several benchmarks for the unified testing and evaluation of neural models for binary analysis, and that this initiative could foster the creation of general multi-tasks architectures that are the equivalent of BERT for our domain.

## Contributions

Our main contribution is the release of BinBench, a benchmark comprising a dataset and several binary analysis tasks. The dataset contains both binary data and a representation of each function in a json format. This latter representation includes key information about each function (assembly instructions, bytecode, arguments, *etc*). This design choice has been made so that researchers can use the dataset without the need to resort to a disassembler or other similar tools.

BinBench contains the following tasks:

- *Binary Similarity*. Check whether two binary functions are generated from the same source code.
- *Function Search*. Given a certain binary function, find its *K* top similar functions in a database.
- *Compiler Provenance*. Identify the compiler family used for generating a certain binary function.
- *Function Naming*. Given a binary function, assign to it a name that matches its semantics and role.
- *Signature Recovery*. Predict the parameters of a given binary function.

BinBench is split in two parts: a blind and a non-blind one. The non blind part contains for each task the labels that a perfect solution is expected to predict. This part can be used to train, validate and test a model. The blind part contains only data without labels. This part must be used for competitive comparison of different models. The blind dataset is composed by binary code taken from several different opensource packages, selected to cover different behaviours of a software system (*e.g.*, it contains networking applications, databases applications, archive management applications, and so on).

The researcher has to use its model to predict a set of labels for the blind dataset, and then submit them to a challenge on EvalAI that will return a performance score without revealing the actual labels (this avoid accidental or intentional overfitting of a model). EvalAI keeps a leaderboard of the best solutions. Its functioning is further explained in the dedicated section.

Finally, for each task we provide a baseline result obtained by running state-of-the-art solutions. The baseline value can be used by researchers as a starting point to check how well their DNNs work against other systems.

# RELATED WORK

## Benchmarks

Deep neural networks have shown state of the art performances in a variety of tasks. One of the fields where these methods shine is the one of natural language processing. It is usual in the NLP community to benchmarks that evaluate the overall quality of model representations. For generality purposes, these benchmarks include multiple tasks on which the architectures can be tested. The tasks are heterogeneous so that they will test different capabilities of a model (*DeYoung et al., 2019*; *Khanuja et al., 2020*).

A recent trend in the field of binary analysis with machine learning technique is the one of using techniques borrowed from the NLP area (*Massarelli et al., 2019b*; *Xu et al., 2017*; *Artuso et al., 2021*; *Liu et al., 2018*; *Massarelli et al., 2019a*; *Rosenblum, Miller & Zhu, 2011*; *Ding, Fung & Charland, 2019*; *Chua et al., 2017*; *He et al., 2018*; *David, Alon & Yahav, 2020*). Surprisingly, in the case of binary analysis community, there is no multi-task benchmark that is commonly adopted. An attempt in this sense is done by BinKit (*Kim et al., 2020*), a benchmark evaluating only the task of binary similarity. Therefore, to the best of our knowledge there is no multi-task benchmark targeting the binary analysis community. Below we detail related work for each task of our benchmark. Furthermore, a comparison of BinBench with relevant datasets is deferred to Discussion section.

## Binary similarity and function search

Binary similarity and function search are related. The first task asks to find whether two functions are similar, accordingly to some definition of similarity, and, usually, to return a similarity score. The latter, instead, asks, given a certain query function, to find the $K$ most similar functions to the query from a certain database. To solve this problem, we may use a binary similarity systems that computes the pairwise similarity between the query and the functions present in the databases, returning the $K$ most similar. For such a reason function search is usually an evaluating tasks of articles proposing binary similarity solutions, and the literature of the two problems is essentially the same.

Several works have studied the problem of binary similarity (*Dullien & Rolles, 2005*; *Khoo, Mycroft & Anderson, 2013*; *Alrabaee et al., 2015*; *David, Partush & Yahav, 2016*; *David, Partush & Yahav, 2017*; *David & Yahav, 2014*; *Lakhotia, Preda & Giacobazzi, 2013*). We will focus on the works that use deep neural networks and machine learning techniques, or that release their datasets.

*Kim et al. (2020)* proposed a public dataset for the binary similarity task. The article uses an approach based on manually selected numeric features. The binary similarity between two functions is computed as an average of the relative differences between selected features. Another public dataset is the one of α from Diff (*Liu et al., 2018*). In this case the article uses manual features, namely the intra-function (raw bytes), the inter-function (function calls) and the inter-module (library imports). The dataset used is a collection of cross-version binaries from the x86 Linux platform. A different approach is proposed in SAFE (*Massarelli et al., 2019b*), which solves the binary similarity task with a self-attentive neural network. The model maps assembly instructions into vectors, using Word2vec from

*Mikolov et al. (2013)*. Then, a self-attentive neural network maps a sequence of instructions into the final embedding of the function. The model has been used to solve several tasks, such as binary similarity, function search, vulnerability search, semantic classification and APT classification. Another article that uses neural networks to transform functions in vectors is Asm2Vec (*Ding, Fung & Charland, 2019*). They use the PV-DM model on a set of random walks performed on the control flow graph (CFG) of a function. In detail, a CFG is directed graph, showing all possible paths of execution for a function. Nodes represent basic instruction blocks, that are sets of consecutive instructions ending with jump or return statement. Each edge, instead, represent a jump from the current instruction block to its successor.

The problem of the binary similarity has also been studied in Gemini (*Xu et al., 2017*). In this case, each binary function is transformed into an attributed control flow graph (ACFG), that is a CFG with manually annotated features. A graph neural network (*Ribeiro, Saverese & Figueiredo, 2017*) is used to map the ACFG into a vector. The similarity between functions is computed as the distance between their vectors.

## Function naming

A recent series of articles is studying the problem of function naming (*Patrick-Evans, Cavallaro & Kinder, 2020*; *Patrick-Evans, Dannehl & Kinder, 2021*; *Gao et al., 2021*). The majority of which are using deep neural networks. For example, *Artuso et al. (2021)* compares the performance of a Seq2Seq (*Bahdanau, Cho & Bengio, 2015*) architecture and a transformer (*Vaswani et al., 2017*). The analysis is carried out on a large set of stripped binaries, built from scratch and publicly available. The function naming problem has also been studied in NERO (*David, Alon & Yahav, 2020*). In this case each call in the CFG is converted to a symbolic call site structure, which includes information regarding the function call and its arguments. From these call site and the CFG an augmented call site graph is created, this graph has the call sites as nodes. The edges of the graph represent potential execution paths of the procedure. Using this representation several neural architectures are tested and compared. A different approach is proposed by Debin (*He et al., 2018*), which predicts debug information from stripped binaries. The assembly code is first translated in BAP-IR. This is an intermediate representation, showing code semantics in an architecture independent way. Therefore, Debin builds a graph representing dependencies among code elements of BAP-IR. Then a conditional random field model is run on top of this graph to predict the missing debug information.

## Signature recovery

Less attention has been devoted to the signature recovery task. The reference article in this case is *Chua et al. (2017)*, that solves signature recovery through neural networks. The model constructs an embedding vector for each instruction through the skip-gram model. To recover the function signatures, RNNs are trained on the resulting instruction vectors to predict types and number of arguments for the functions.

A more general problem is the one of generating an high level representation from binary code, decompilation process. Many works have tackled this problem, with and

without neural networks, *Katz, Ruchti & Schulte (2018)*, *Katz et al. (2019)*, *Fu et al. (2019)*. However, our task focuses on just retrieving the signature of the function. A related problem is the one of finding low level patterns in binaries (*Escalada, Ortin & Scully, 2017*; *Escalada, Scully & Ortin, 2021*) as these can be used to aid the recovery of input and return types of a function.

### Compiler provenance

The task of compiler provenance has been studied in several articles. For example, *Rosenblum, Miller & Zhu (2010)* represents a binary program as sequence of instructions (idioms). This kind of representation removes details such as memory offsets and literals. In *Rosenblum, Miller & Zhu (2011)*, instead, the task of compiler provenance is evaluated as a classification problem. This approach automatically selects features from idioms and subgraphs of the CFG (graphlets). Therefore, each function is associated to a binary vector, representing the presence of each feature. The model is implemented using an SVM classifier. The approach of *Rahimian et al. (2015)*, instead, extracts a set of syntactic features and compiler tags from known compiled code. Therefore, it uses ACFGs to extract semantics features from binaries. The solution allows to identify the compiler provenance and the optimization level. In *Chen et al. (2019)*, the task of compiler provenance is solved through neural networks. First, the architecture represents an input binary as a list of functions. Therefore, each function is converted into vector and the word embedding is applied to the instructions. Finally, the model retrieves compiler information using recurrent neural networks.

In *Massarelli et al. (2019a)*, features are automatically extracted from the CFG. The model associates each graph vertex to a vector representation. Therefore, the graph embedding results from the aggregation of the vertex vectors. The approach evaluates two tasks, namely binary similarity and compiler provenance.

## TASKS

For BinBench we selected the following tasks: binary similarity, function search, compiler provenance, function naming and signature recovery. These widely different tasks are meant to capture distinct aspects of a binary function, and thus to assess the generality of the architecture under test. All these tasks already appeared in the literature (see the related work section for an in depth discussion). We decided to use known tasks for several reasons: we want tasks that have real-world relevance and thus could be used to assess the practical usability of a DNN architecture; we preferred well known tasks as they have already been accepted by the community, and this could improve the acceptance of our benchmark; furthermore, by using known tasks we can readily find solutions that can be used to create baseline scores for the benchmark. Roughly, we can divide our tasks in two sets: semantic tasks and syntactic tasks. Semantic tasks require the network to abstract from the syntactical representation of the code and to learn semantic aspects of the binaries; in this category there are the binary similarity and function search tasks where functions compiled from the same source code are recognised and the function naming task, which requires enough understanding to give a meaningful name to a snippet of

binary code. The syntactic tasks, in a dual way, require to focus on the syntactic aspect of binary; in this category we have the compiler provenance task, which recognises the compiler used to generate a certain binary, and the signature recovery task, that requires to understand how many and which kind of arguments a function is taking. In the following we detail each task.

## Binary similarity

In this task the network has to understand if two binary functions are based on the same source code or not. The binary similarity is far from trivial, as, it is well known (*Xu et al., 2017*) that different compilers and optimisation levels are able to generate markedly different binary code. This problem has been extensively investigated in the last years (*Haq & Caballero, 2021*). We decided to use the formalisation first used in *Ding, Fung & Charland (2019)* and then also in *Massarelli et al. (2019b)*, where the problem is expressed over pairs of functions. This task has a practical importance in several applications: clone search, copyright infringement dispute, *etc*. It is a semantic task as it requires the network to abstract from the syntactic difference created by the compiler.

### Formal definition
Two binary functions $f_1, f_2$ are similar, $f_1 \sim f_2$, if they are the results of compiling the same original source code $s$ with different compilers. Essentially, a compiler $c$ is a deterministic transformation that maps a source code $s$ to a corresponding binary function $f^s$. We considered as a compiler the specific software, *e.g.*, gcc-5.4.0, together with the parameters that influence the compiling process, *e.g.*, the optimization flags $-O[0, \ldots, 3]$. In this task we provide a set of unlabelled binary function pairs $p_1, \ldots, p_n$, the tested architecture has to classify each pair as similar (label $+1$) or dissimilar (label $-1$). Examples of binary similarity are showed in Fig. 1.

### Metrics
We evaluate the quality of the solution using the Area Under the Curve (AUC) considering the classification task as explained above. The AUC, as the name suggests, is the area under the receiver operator characteristic (ROC) curve, which represents the relation between true positive rate (TPR) and false positive rate (FPR). Therefore, the AUC shows the overall classification capabilities of an observed model (*James et al., 2013*).

## Function search

The function search task similar to the binary similarity one. Also in this case the goal is to find functions that are similar. The main difference is that the previous task is a classification task where pairs of functions are labelled as similar or not; conversely, the function search is a retrieval task. In this case a function, used as *query*, has to be looked up on a *database* of other functions. The query has to return the $K = 20$ most similar functions. The value assigned to $K$ has been chosen with respect to $S$, the average number of similar functions available in the complete dataset, which is slightly greater than $K$. This task models a real use case of the binary similarity search where a certain sensitive

**Peer**J Computer Science

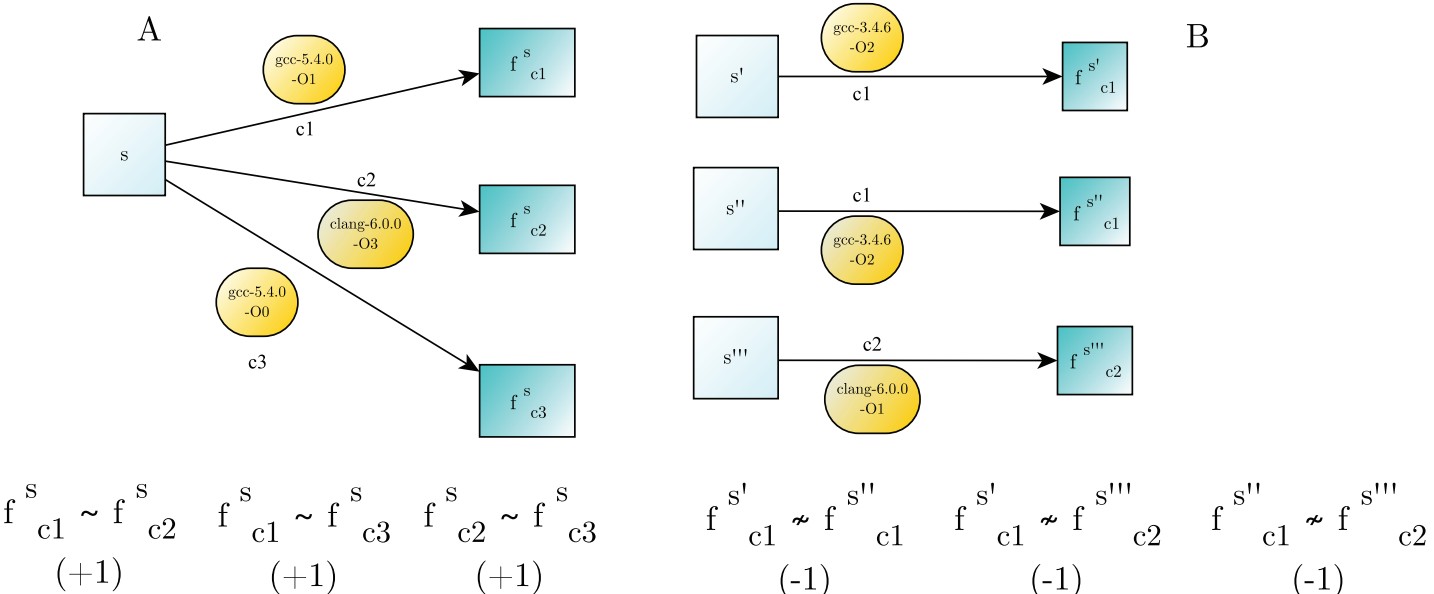

**Figure 1 Examples of binary similarity.** (A) Similar binary functions are generated by compiling the same source code. (B) Dissimilar binary functions are obtained from the compilation of different source code.

function, that could represent a piece of a malware or part of copyrighted software, has to be searched on a database.

### Formal definition

We have a database *DB*, that can be seen as a set of *n* different binary functions and a set of queries $Q : \{f_1, f_2, \ldots, f_q\}$. In the function search task for each function $f_i \in Q$, the networks has to return a set $A_j \subset DB$ of size $K$. The functions in $A_j$ have to be similar to function $f_i$.

### Metrics

The quality of the solution is determined using precision, recall, F1 score and normalized discounted cumulative gain (nDCG). The evaluation is computed on the $K = 20$ most similar functions proposed in the solution. In detail, the *precision* represents the fraction of correctly classified samples over the total retrieved samples. Therefore, given the true positives (TP) and the false positives (FP), the precision is computed as follows.

$$Precision = \frac{TP}{TP + FP}$$

The *recall*, instead, measures the number of correctly retrieved instances over the total number of instances of the correct answer.

$$Recall = \frac{TP}{TP + FN}$$

where FN represents the false negatives (FN). The *F1 Score* shows the overall precision of a model and it is computed as the harmonic mean of precision and recall.

$$F1 - Score = 2 * \frac{Precision * Recall}{Precision + Recall}$$

Finally, the *nDCG* is the measure of the ranking quality for retrieval tasks. Given a binary function (query), our goal is to retrieve its $K$ most similar functions from a database. The returned solution should have the similar functions in first positions. As an example, consider the optimal query answering $s_{optimal} = (f_1, f_2, f_3)$, where each $f_{i\ (i=1,\dots,3)}$ is similar to the query $f_q$. Now suppose to have two models $m_1, m_2$ that return the following solutions: $s_{m_1} = (f_4, f_1, f_2)$ and $s_{m_2} = (f_1, f_4, f_2)$, where $f_4$ is not similar to $f_q$. Both solutions include the same set of functions, but $s_{m_2}$ is better that $s_{m_1}$. This because $m_1$ places the a non similar function at the beginning of the list. Formally, the nDCG is defined as

$$nDCG(R_{\vec{f}}) = \frac{\sum_{k=1}^{k} \frac{Similar(r_i, \vec{f})}{log(1+i)}}{IdealDCG_k}$$

where:

– $f$ is the query function.
– $R_{\vec{f}} : (r_1, r_2, \dots, r_k)$ are the top-k similar functions.
– $Similar(r_i, f)$ is equal to 1 if $r_i$ is similar to $f$, 0 otherwise.
– $IdealDCG_k$ is the discounted cumulative gain of the optimal query answering.

The nDCG has a value between 0 and 1, and it depends on the *order* given to the returned similar functions. Therefore, the nDCG is greater when similar functions are the first elements of the solution. Consider the example discussed above. The optimal query answering is $s_{optimal} = (1, 1, 1)$, while two model solutions are $s_{m_1} = (0, 1, 1)$, $s_{m2} = (1, 0, 1)$. Each element in the solutions is 1 if the returned function is similar to the query, and 0 otherwise. Therefore, $s_{m_2}$ is a better solution than $s_{m_1}$, and $nDCG_{s_{m_2}}$ is greater than $nDCG_{s_{m_1}}$.

## Compiler provenance

In the compiler provenance task, given a binary function the goal is to classify it accordingly to the compiler and/or optimization level that generated it. This problem was proposed for the first time in *Rosenblum, Miller & Zhu (2010)*, and it has then been studied in several articles (*Rosenblum, Miller & Zhu, 2011*; *Rahimian et al., 2015*; *Chen et al., 2019*; *Massarelli et al., 2019a*). Knowing the compiler that has generated a certain binary is needed to use specific library detection toolkits (such as IDA FLIRT; https://hex-rays.com/products/ida/tech/flirt/in_depth/); another example is *Caliskan et al. (2018)* where this information is necessary to understand who is the writer behind a certain snippet of binary code. This is a syntactic task as a good solution has to learn the syntactic structure generated by a certain compiler without considering the semantic of the code.

### *Formal definition*

We have a set $C: \{c_1, c_2, \ldots\}$ of possible compilers, these compilers are divided in families $F_1, \ldots, F_m$ a family is a specific compiler without considering the version and optimization level (*e.g.*, the clang family contains all the versions of clang). In the compiler provenance task we are given a set of functions $f_1, \ldots, f_n$ and we have to label each of them with the correct family; *i.e.*, given $f_j$ compiled with compiler $c_j$ we have to output the family $F$ such that $c_j \in F$.

### *Metrics*

We evaluate the compiler provenance task using four metrics: accuracy, precision, recall, and F1 score. In detail, the *accuracy* of the model is computed as follows.

$$Accuracy = \frac{TP + TN}{TP + FP + TN + FN}$$

where TN represents the true negatives.

## Function naming

In this task the purpose is to predict an appropriate name for a binary function. This problem was introduced in *He et al. (2018)* and it has been then studied in *David, Alon & Yahav (2020)*, *Artuso et al. (2021)*. This naming task has a practical importance, it could helps an analysts that is reversing an unknown binary by identifying key functionalities; as example, encryption or networking functions. Function naming is a semantic task as it requires the network to learn the semantic of a function translating it into a description in natural language that constitutes a name. A key problem is the evaluation of function naming solutions as different programmers could use widely different naming conventions. An approach proposed in the literature is to split names in substrings (*Patrick-Evans, Dannehl & Kinder, 2021*), more details in the metrics section.

### *Formal definition*

Given a vocabulary $V$ and a set of functions $f_1, f_2, \ldots, f_n$ in the function naming problem we have to assign to each function $f_j$ a string $s_j$ composed by words in the vocabulary $V$. The string $s_j$ has to be *"meaningful"*, it has to capture the semantic of the function and its role inside the software (*Artuso et al., 2021*).

### *Metrics*

To measure the quality of the solution for the function naming task, we consider the following metrics: precision, recall and F1-score and BLEU.

To be evaluated, each function name is represented as a list of tokens. In detail, function names are first split on underscores, and then, English word segmentation (https://grantjenks.com/docs/wordsegment/) is applied. This tool split strings into "meaningful" English words. For example, we consider the following function names: `setValue`, `set_value`, and `setvalue`. Even if they have different naming convention, they are composed by the same English words. Therefore, splitting these function names generates the same output: `["set", "value"]`.

For the labeled dataset, we provide two different labels: the original name, and the one split with our technique. We believe that having both labels can be helpful to train models. For the blind dataset, the user has to provide a name composed by the tokens of the vocabolary (https://github.com/grantjenks/python-wordsegment/tree/master/wordsegment) used in the split procedure. This list of tokens will be compared with the list of correct labels obtained by applying the split procedure to the original function name.

In detail, for each function name we consider:

- Two list of tokens: correct labels $l$ and predicted labels $p$,
- A function $x_a$, such that $x_a = 1$ if $p_a \in l$ and 0 otherwise ($\forall 0 < a \leq |p|$)
- The score of a prediction $score_p = \sum_{k=1}^{|p|} x_a$

Therefore, *individual metrics* are computed as:

$$Precision = score_p / |p|$$
$$Recall = score_p / |l|$$
$$F1 - Score = 2 * (Precision * Recall) / (Precision + Recall)$$

Finally, we individually evaluate each prediction with BLEU (*Papineni et al., 2002*), which measures the quality of machine translations. It is language independent and it returns a value between 0 and 1. In detail, BLEU score is defined as follows.

Given the length of the prediction $c$, and the effective length $r$, the brevity penalty (BP) is

$$BP = \begin{cases} 1 & (\text{if } c > r) \\ e^{(1-r/c)} & (\text{if } c <= r) \end{cases} \tag{1}$$

Therefore, BLEU score is defined as

$$BLEU = BP \cdot exp\left(\sum_{n=1}^{N} w_n log p_n\right)$$

where $N$ is the maximum n-gram length, $p_n$ is the modified precision score, and $w_n$ represents corresponding weights.

The *returned metrics* represent performances over all the predictions. They are computed as average of each individual measure.

## Signature recovery

Given a binary function the task consists in predicting its parameters, that is the number arguments it takes and the type of each one. The signature recovery was introduced in *Chua et al. (2017)*, and it has been solved, as a subproblem, also in *He et al. (2018)*. As noted by *Chua et al. (2017)*, the signature recovery task is useful for control-flow hardening (*Zhang & Sekar, 2015*) and taint-tracking (*Saxena, Sekar & Puranik, 2008*). This is a syntactic task, the network has to learn how the compiler handles the parameters in the prologue of a function. For this task, we have selected the data types used by *Chua et al. (2017)*. We have also included void, that should be intended as absence of parameters.

### Formal definition

We have a set of types $T$: {`pointer, enum, struct, char, int, foat, union, void`} and a set of functions $f_1, f_2, \ldots, f_n$. In the signature recovery task given as input a function $f_j$ the network has to output a multiset $P_j$ with elements in $T$; this multiset represents the parameters that the function takes with their type. The prediction is a multiset to model functions having multiple parameters of the same type; *e.g.*, if function sum takes two integers the correct prediction is {`int,int`}.

To prevent misinterpretation of data types by the decompiler, we have chosen a subset of all the data types available, while excluding the others. This is a representative subset, as it is composed by the most various data types.

In detail, we have included *int*, because it can be used to represent both integers and boolean values (as 0 and 1). We have inserted *pointer*, which is used to perform memory accesses. We have also selected *char*, *float* and *void* data types to represent, respectively, non numeric symbols, floating numbers and absence of parameters. Finally, to represent complex data types, we have included *enum*, *struct* and *union*.

### Metrics

The solutions proposed for the signature recovery task are evaluated using the accuracy, the precision, and the recall. The metrics are computed using the micro averaging method, which allows to properly evaluate a multiclass classification problem in unbalanced datasets. This method, however, produces metrics with same values (*Herrera et al., 2016*).

## DATA SOURCE

We select 131 packages from the core repository of Arch Linux (https://archlinux.org/packages/), which is a Linux distribution optimized for x86-64 architectures. We build each package using the PKGBUILD shell script used by the Arch Linux package manager. These files are needed to compile software packages with makepkg, a compile script included in the Arch Linux package manager.

We choose two compilers that are commonly used for research purposes, namely GCC and Clang. To balance the total number of resulting binaries, five different versions for each compiler have been considered. The selected compilers are the following: gcc versions 6, 7, 8, 9, 10, and, clang versions 4, 6, 8, 10, 11.

During compilation we keep debug symbols (−g) and we compile the packages with four different optimization levels: −O0, −O1, −O2, −O3. We transform the binary function in textual data formatted using JSON. In our training split, we provide both JSON and corresponding binaries. These compiled files can be helpful for custom training. It should be noted, however, that provided binaries are not stripped. Therefore, it is recommended to strip files before using them as model input as they could contain debug information that could unintendedly misguide a model that does not want to use such additional information.

We discard a package if the compile process fails, or when the binaries produced by different optimization levels are equal; this last case is to avoid the introduction of duplicated functions in the dataset. Duplicate removal is performed in order to prevent

biases during evaluation, as done also in *David, Alon & Yahav (2020)*, *Massarelli et al. (2019b)*, *Chua et al. (2017)*.

## DATASET

To compile the selected packages, we set up docker containers of Arch Linux and use parallel processing. The compilation process generates 1,127,479 binaries in total. Arch linux is a from scratch distribution that can be configured to compile packages with our compiler and optimization level of choice taking care of packet dependencies with the packet manager. From each compiled file, we extract a set of functions to be included in BinBench. The disassembled assembly and CFGs are retrieved using Ghidra (https://github.com/NationalSecurityAgency/ghidra), we remark that we do not use a symbolic or dynamic approach, therefore we do not predict the target of indirect jumps and calls. This choice is also done in *Xu et al. (2017)*, *Ding, Fung & Charland (2019)*, *Massarelli et al. (2019b)*. We use Pyelftools APIs to obtain the function signatures information (https://github.com/eliben/pyelftools). The textual data are are stored in individual JSON files, one for each binary function. The total number of source code functions is roughly 132,000, this gives 4,408,191 different binary functions. The average number of similar functions is $S = 26$.

We split the binary functions in two sets, namely *labeled dataset* and *blind dataset*. The former is intended for the training process, while the latter is meant for testing. The blind split contains manually selected open source packages. The packages have been selected so that they include applications of different kind that can cover the majority of behaviours in software systems. For example, we have included packages for network communication (*e.g.*, OpenSSL), database management (*e.g.*, SQLite), shell scripting (*e.g.*, Bash) and archive management (*e.g.*, Tar).

The labeled dataset has two components. The first is the JSON component, a set of binary functions in JSON format. The second is the binary component, which includes compiled files from which the functions are extracted.

In detail, each JSON file in the labeled dataset represents a function with the following fields:

- *asm*: instructions composing the function, represented in assembly and bytecode formats. Each instruction block is linked to its offset,
- *called*: callee functions,
- *callers*: caller functions,
- *cfg*: CFG of the function. It is a directed graph. It is represented as a list of edges, defined as pair source and destination blocks and list of blocks composing the graph (for each block it is possible to retrieve the list of assembly instructions and the bytecode).
- *compiler*: compiler version used to produce the binary,
- *name*: the function name retrieved using the debug information,
- *opt_level*: optimization level used during the compilation,
- *origin_file*: name of the binary file from which the function has been extracted,
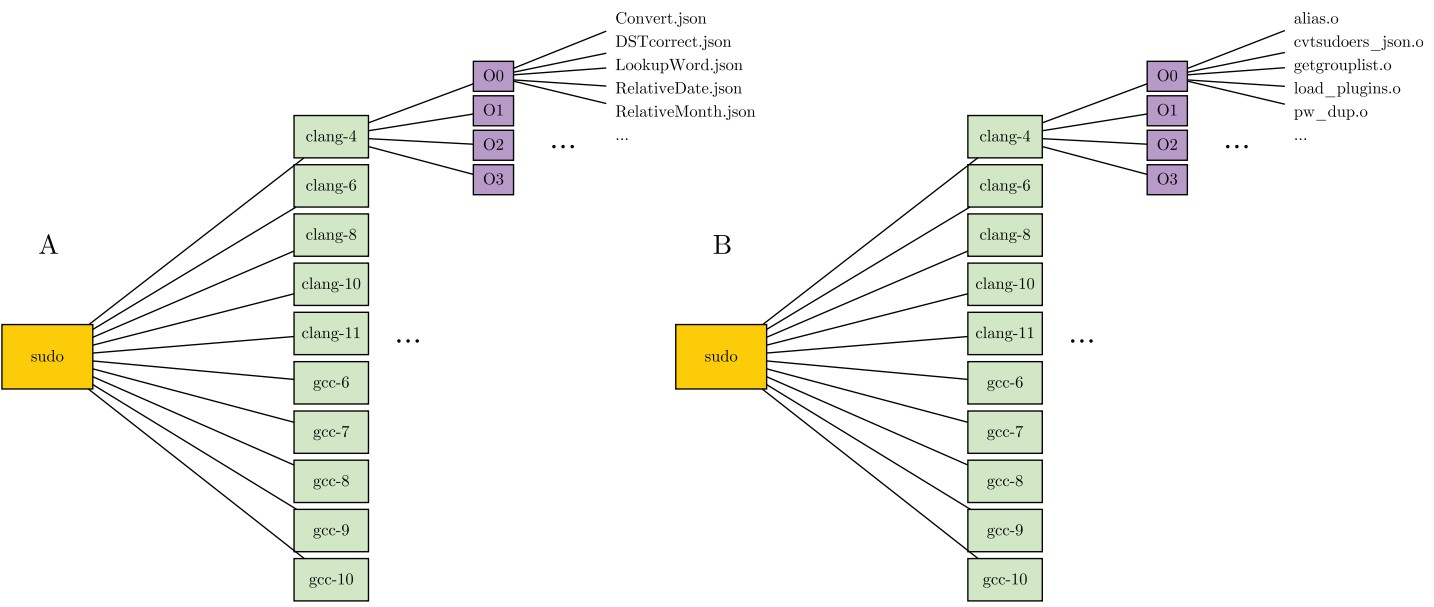

**Figure 2 Folder structure of a package in the labeled dataset.** (A) Folder in the JSON component. (B) Folder in the binary component.

- *package*: name of the package of the origin_file,
- *parameters*: list of parameters for the function. Each parameter is represented with its name and type. Types can be one of the following: pointer, float, char, int, enum, struct, union, void (the function takes no parameters).
- *return_type*: the return type of the function.

The two components of labeled dataset are specularly organized in folders and stored separately. In particular, packages are represented as parent folders, named with the package's name. Furthermore, each package contains several subfolders, one for every compiler version. In turns, each compiler folder contains a set of subfolders, one for each optimization level. The optimization level folders are used to store the actual dataset files. In detail, the JSON component stores functions in JSON format, named with corresponding function names. The binary component, instead, stores binary files, named with names of the compiled files. Figure 2 shows the structure a package folder in the labeled dataset.

The blind dataset is the one used for EvalAI (explained in detail in the dedicated section). This dataset only includes binary functions represented in JSON format. In this case, function names are hidden. For this reason, each JSON function is named with a unique ID instead of using its name. Furthermore, we have removed duplicate functions of the blind dataset. Two functions are duplicated if they contains the same sequence of instructions without considering memory offsets and immediate values. Any function that is equal to another one is considered as a duplicate and removed. Finally, we stored resulting JSON files in a single folder.

**Table 1 Tasks and datapoints contained in each dataset split.**

| Task name | Labeled dataset | Blind dataset |
|---|---|---|
| Binary similarity | 732,376 | 243,044 |
| Compiler provenance | 89,744 | 9,600 |
| Function naming | 120,640 | 9,600 |
| Signature recovery | 120,259 | 9,086 |

**Table 2 Number of datapoints for the function search task.**

| Dataset | Query | Database |
|---|---|---|
| Labeled | 30,000 | 600,000 |
| Blind | 10,000 | 200,000 |

In the the blind dataset, each JSON file includes only a subset of the fields given in the training set. In detail, a function in the blind dataset is represented by: *asm*, *called*, *callers*, *cfg*.

For each task we have selected a set of datapoints, that are the predictions to be computed by the model. In detail, a datapoint is a couple containing an entry in the dataset and the label to be predicted on that entry (which depends on the specific task). The datapoints are stored in individual JSONL files, one for each task in both datasets. In the labeled dataset, JSONL files contain solutions for the predictions (*i.e.*, label fields of the datapoints are filled). In the blind dataset, instead, JSONL files do not contain any solution (*i.e.*, label in the datapoints are empty). These labels are the one that should be predicted for solving EvalAI tasks. Therefore, they have to be filled by the model itself and sent back to the benchmark.

The Table 1 shows the datapoints that have to be evaluated for each task. The Table 2, instead, shows the number of datapoints of Function Search task. In this case, the datapoints are splitted in two sets. The first one contains the queries, that are the functions for which the model has to find their $K$ similars. The second set is the database, in which the $K$ similar functions have to be found.

We point out that these datapoints are using a subset of the functions included in the whole dataset. Each function in BinBench can be used to train/test an architecture on all the proposed tasks.

## EVALAI

To host our benchmark, we choose EvalAI (https://eval.ai/). This is an open-source platform for performance testing of machine learning algorithms. In EvalAI, each user can upload his own challenge and share it with the community. Furthermore, each challenge can be divided in multiple phases. Thanks to this functionality, we are able to host our benchmark. Therefore, we create the BinBench challenge, where each task is an EvalAI phase.

**Table 3 Results for the function search task.**

| Precision | Recall | F1 score | nDCG score |
|---|---|---|---|
| 0.26 | 0.26 | 0.26 | 0.36 |

For each task, we create two versions of the datapoint file, both in JSONL format and containing the same datapoints. The first one is intended for the parteicipants and therefore it contains the unlabeled datapoints. The predicted labels are inserted in this file and then sent back to BinBench to be evaluated. The second file contains the labeled version of the datapoints. It is used as ground truth to evaluate parteicipant solutions. Each entry for the challenge is uploaded on BinBench and then compared with the correct solution. Depending on the task being solved, different metrics will be returned. This evaluation allows to check the performances of a model.

## BASELINES

To generalize our analysis, we want to evaluate our benchmark with various architectures, trained on different datasets. Therefore, whenever possible we use existing pretrained models as baselines.

### Binary similarity baseline

For binary similarity and function search, we use the pretrained model of SAFE (https://github.com/gadiluna/SAFE), implemented with Tensorflow and Python. SAFE computes an embedding in two phases. First, it embeds each assembly instruction using word2vec model (*Mikolov et al., 2013*). Therefore, it computes the final embedding using a *Self-Attentive Neural Network*. In detail, this network is a bi-directional RNN that produces a summary vector for each input instruction, and then it computes the function embedding as a weighted sum of all summary vectors. We embed each binary function using SAFE. Therefore, we compute the cosine similarity between requested couples. We consider a predefined threshold $T = 0.6$ to convert each cosine similarity into a label. In detail, a couple has been marked as similar (label +1), when the cosine similarity is greater than $T$, and dissimilar (label $-1$) otherwise. This baseline achieves an AUC = 0.91 on the binary similarity task.

### Function search baseline

To solve function search task, we use the SAFE pretrained model described previously. In detail, we first embed every binary function of the query list and the database. Therefore, we compute binary similarity measure between each embedded query, and each embedded database function. Finally, we analyze the resulting similarities. For each query, we retrieve the $K$ most similar database functions (which have greater value of the computed measure). The outcomes are ordered with respect to the grade of similarity (*i.e.*, the first function is the most similar to $q$, the second is the second most similar and so on). This ordering is respected for each function in the query list.

The results for the function search task are reported in Table 3.

**Table 4 Results for the compiler provenance task.**

| Accuracy | Precision | Recall | F1 score |
|---|---|---|---|
| 0.81 | 0.78 | 0.88 | 0.82 |

## Compiler provenance baseline

For compiler provenance, we use a pretrained model (https://github.com/lucamassarelli/Unsupervised-Features-Learning-For-Binary-Similarity) of the graph embedding neural network described in *Massarelli et al. (2019a)*. The model embeds a CFG graph through two components. The first one is the *Vertex Feature Extraction*, that maps each vertex of the CFG into a feature vector. The second component, the *Structure2Vec network*, uses deep neural networks to produce the final graph embedding. This component creates a vector for every graph vertex. In the training phase, those vectors are dynamically updated using an approach based on rounds. Vector updates take into account graph topology and previously extracted features. Therefore, the graph embedding is computed as the aggregation of updated vectors. The model is implemented using Tensorflow and Python. To solve our task, we have first trained the model on its own dataset, and then we have used it to infer requested datapoints. In detail, the dataset used for the training is the *restricted compiler dataset* (https://github.com/lucamassarelli/Unsupervised-Features-Learning-For-Binary-Similarity), which includes several open-source projects compiled. The packages are compiled for AMD64 using three different compilers (gcc 3.4, gcc 5.0 and clang 3.9) and four optimization levels. Therefore, code has been disassembled using radare2 (https://rada.re/). The results are showed in Table 4.

## Function naming baseline

We use pre-trained transformer of in nomine function (https://github.com/gadiluna/in_nomine_function) to solve function naming. In detail, to retrieve names, the solution represents binary functions as list of normalized instructions. Furthermore, each function name is transformed into tokens. To solve the task, two architectures are used, namely Seq2Seq and Transformer. They are trained over a big dataset for a maximum of 30 epochs with early stopping mechanism, Adam optimizer and batch size of 512. The model has been implemented using OpenNMT-py. We evaluate our dataset with the pre-trained Transformer. However, the obtain performances are lower with respect to the other tasks. This may be caused from the fact that our dataset includes tokens on which the model was not trained. Results are reported in Table 5

## Signature recovery baseline

To solve signature recovery task, we have trained a Transformer from scratch through OpenNMT-TF (https://github.com/OpenNMT/OpenNMT-tf). We have used the default configuration of the architecture, and trained it over a portion of our training dataset. In detail, the training set were composed by 115,000 randomly picked binaries, where 15,000 were used for the evaluation phase. During the training, we have used a batch size of 200,

**Table 5 Results for the function naming task.**

| Precision | Recall | F1 score | BLEU |
|---|---|---|---|
| 0.07 | 0.04 | 0.05 | 0.03 |

**Table 6 Results for the compiler provenance task.**

| Accuracy | Precision | Recall |
|---|---|---|
| 0.53 | 0.53 | 0.53 |

**Table 7 Comparison with datasets available in literature.**

| Dataset | Number of functions | Evaluated task | Open source |
|---|---|---|---|
| BinKit *Kim et al. (2020)* | 75,230,573 | ① | Yes |
| In nomine function *Artuso et al. (2021)* | 8,861,407 | ④ | Yes |
| α Diff *Liu et al. (2018)* | 4,979,586 | ① | Yes |
| BinBench | 4,408,191 | ①, ②, ③, ④, ⑤ | Yes |
| SAFE *Massarelli et al. (2019b)* | 548,133 ①, 581,640 ②, 1,587,648 ③ | ①, ②, ③ | Yes |
| Graph embedding NNs *Massarelli et al. (2019a)* | 95,535 ①, 2,040,246 ③ | ①, ③ | Yes |
| Toolchain provenance *Rosenblum, Miller & Zhu (2011)* | 955,000 | ③ | No |
| Asm2Vec *Ding, Fung & Charland (2019)* | 139,936 | ② | No |
| Gemini *Xu et al. (2017)* | 129,365 | ① | No |
| Eklavya *Chua et al. (2017)* | 119,352 | ⑤ | Yes |
| NERO *David, Alon & Yahav (2020)* | 67,246 | ④ | Yes |
| Debin *He et al. (2018)* | 238 | ④ | No |

**Note:**
Evaluated tasks: ①, binary similarity; ②, function search; ③, compiler provenance; ④, function naming; ⑤, signature recovery.

adam optimizer. We have evaluated our model every 100 steps stopping on the highest result on the validation dataset. The highest performances has been observed from the model that was trained in 3,850 steps. We remark the fact that metrics are computed using the micro averaging method. The results are showed in Table 6.

## DISCUSSION

### Comparison with other datasets

We compare BinBench with the available literature. For a fair comparison, we have reported the number of functions for each of the tasks included in BinBench, excluding the dataset parts used for other tasks. In Table 7 we report the results of our comparison, that we discuss below.

- **Binkit *Kim et al. (2020)*** proposes a dataset for the binary similarity task. The dataset is obtained by compiling 51 GNU packages using nine different compilers and five

optimization levels. The dataset is divided in subsets, one for each different compilation options.

- **In Nomine Function** *Artuso et al. (2021)* uses AMD64 packages retrieved from Ubuntu 19.04 repository. Duplicates functions were removed as well as functions for which symbolic names cannot be retrieved. The dataset is built for the function naming problem.

- α **Diff** *Liu et al. (2018)* uses a dataset composed by two different collections of source codes. For each of the projects included in the dataset, different versions have been retrieved. The first collection is composed by 31 projects, compiled using GCC v5.4. The second collection is composed by the binaries of 895 packages from Debian repository.

- **SAFE** *Massarelli et al. (2019b)* Proposes two datasets for the binary similarity task (one for AMD64 and another for ARM) and one dataset for the function search task. The first training set is obtained by nine projects for AMD64, compiled with three compilers and four optimization levels. The second dataset is composed by two versions of Openssl, compiled for ARM with GCC v5.4 and four optimization levels. For the task of function search, the dataset is composed by packages of AMD64, compiled with 10 compilers and four optimization levels.

- **Graph Embedding NNs** *Massarelli et al. (2019a)* proposes two datasets for compiler provenance. The first is composed by open-source projects, compiled with three different compilers and four optimization levels, while the second is constituted by several projects, compiled with 11 compilers.

- **Toolchain Provenance** *Rosenblum, Miller & Zhu (2011)* solves the task of compiler provenance on a dataset composed by eight open-source packages, compiled with nine versions of three compilers, using two optimization options. The dataset is not released.

- **Asm2Vec** *Ding, Fung & Charland (2019)* proposes two different datasets for the function search task. The first one is composed by binaries of 10 libraries, generated with GCC v5.4 and four optimization levels. The second dataset is composed by a subset of the previous one, composed by four libraries. It is built using Clang and four obfuscation options of Obfuscator-LLVM. The dataset is not distributed.

- **Gemini** *Xu et al. (2017)* solves the binary similarity task. It uses a training dataset, obtained by compiling two versions of OpenSSL, using GCC v5.4 in four optimization levels. The dataset is not released.

- **Eklavya** *Chua et al. (2017)* solves the signature recovery task on a dataset composed by Linux packages, compiled with two compilers and debugging symbols.

- **NERO** *David, Alon & Yahav (2020)* uses a dataset for the function naming task. The dataset is built using packages from GNU repository.

- **Debin** *He et al. (2018)* uses a dataset composed by Linux packages, that is not being released. The packages compiled in multiple optimization levels and different compilers.

We want to point out that BinBench is the only dataset built for the evaluation on all the considered tasks. Furthermore, it provides both binaries and JSON representations of the

evaluated functions. Table 7 shows the number of functions included in each dataset. Moreover, it shows whether or not a dataset is available for downloading.

## CONCLUSIONS

We proposed BinBench a multi-task dataset for comparing deep neural networks solutions for binary functions representation. The dataset has been designed to include several different tasks to evaluate the expressiveness of a certain model. We released our dataset to the public and we evaluated it using baselines for each task we proposed.

## ACKNOWLEDGEMENTS

We want to thank Fabio Petroni for useful discussions in the first phase of this work.

### Funding

Francesca Console is supported by TIM S.p.A. through the PhD scholarship. This work was supported by project SERICS (PE00000014) under the MUR National Recovery and Resilience Plan funded by the European Union—Next Generation EU. The funders had no role in study design, data collection and analysis, decision to publish, or preparation of the manuscript.

### Grant Disclosures

The following grant information was disclosed by the authors:
TIM S.p.A. through the PhD Scholarship.
SERICS: PE00000014.
European Union—Next Generation EU.

### Competing Interests

The authors declare that they have no competing interests.

### Author Contributions

- Francesca Console conceived and designed the experiments, performed the experiments, analyzed the data, performed the computation work, prepared figures and/or tables, authored or reviewed drafts of the article, and approved the final draft.
- Giuseppe D'Aquanno conceived and designed the experiments, performed the experiments, analyzed the data, performed the computation work, authored or reviewed drafts of the article, and approved the final draft.
- Giuseppe Antonio Di Luna conceived and designed the experiments, authored or reviewed drafts of the article, and approved the final draft.
- Leonardo Querzoni conceived and designed the experiments, authored or reviewed drafts of the article, and approved the final draft.

### Data Availability

The code used for creating the dataset is available at GitHub and Zenodo: https://github.com/fr4nc3sc4/BinBench_;

fr4nc3sc4. (2022). fr4nc3sc4/BinBench_: v1.0 (v1.0). Zenodo. https://doi.org/10.5281/zenodo.7296593.

The dataset is available at FigShare: Console, Francesca; Querzoni, Leonardo; Di Luna, Giuseppe; D'Aquanno, Giuseppe (2023): BinBench: Dataset for Binary Function Representations. figshare. Dataset. https://doi.org/10.6084/m9.figshare.21546111.v2.

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
