# Peer review of "BinBench: a benchmark for x64 portable operating system interface binary function representations"

_PeerJ Computer Science, doi:10.7717/peerj-cs.1286_

## Round 0.1 · original submission · Major Revisions

The manuscript is interesting. Reviewers have suggested the revisions both interms of the presentation and technical aspects as well.

·

Excellent Review

This review has been rated excellent by staff (in the top 15% of reviews)
EDITOR COMMENT
A detailed and comprehensive review written by the reviewer with lot of insights into the article, that would certainly help the authors to improve the article.

Basic reporting

SUMMARY

This paper describes a multi-task benchmark for evaluating the performance of machine learning models that work with binary code. This benchmark comprises a dataset and 5 tasks that perform different syntactic and semantic analyses.

As the authors suggest, in the existing literature, most of the papers use their custom datasets and frequently acknowledge problems by trying to compare their solutions. It is reasonable to expect that the adoption of commonly accepted benchmarks will facilitate the comparison between models.

I would like to encourage the authors to continue working on this interesting and valuable problem. It is also my opinion that this type of benchmark will be very useful for the reverse engineering community. However, this paper needs a major revision to deal with the listed concerns.

MAJOR COMMENTS

6. Related to the “Function Search” task, it could be explained in a clearer way, perhaps with a motivating example to help the reader to understand this task. Namely:
a. It is not clear to me why the nDCG score is needed. If a solution of size K is returned with some functions which are truly similar (TPs) and some errors (FPs). Why it is so important to have all the TPs at the “beginning” of the set? The user of this solution is not going to know which ones are TPs or FPs.
b. What do you use as the “Similar” function? Any kind of distance function? I have not found a field in the dataset grouping the functions in clusters based on their similarity.
c. A deeper explanation in the “Function Search Baseline” section would be also advisable.

7. Concerning the Related work:
a. I have noticed the lack of literature related to the “Function Search” task.
b. I would also like to reference some publications that undertake decompilation “as a whole” using a Neural Machine Translation approach: Katz (Deborah) et al, Katz (Omer) et al and Fu et al. They achieve some of the proposed tasks in order to generate the decompiled code.
c. I would also include the papers already mentioned in other parts of this review:
* Escalada et al (2021)
* Cnerator
* Escalada et al (2017)

MINOR COMMENTS

3. In lines 310 to 311, it appears “[We discard a package]… when the binaries produced by different optimization levels are equal; this last case is to avoid the introduction of duplicated functions in the dataset.”. It is unclear to me why this is necessary or even why it is undesirable to have several binaries with the same binary representation even though they were produced with different optimization levels.

4. In the “Compiler Provenance” definition appears a reference to Lafferty et al (2001) as a paper that has studied compiler provenance. This paper is the seminal work on Conditional Random Fields, so it is not appropriate to put it there.

5. In line 249, both references to Rosenblum et al must be exchanged.

6. Considering the following sentences as an example:
a. (Abstract) Line 14: “… first and foremost, the one of making …”
b. (Abstract) Lines 15 to 16: “… is the one of being able to …”
c. (Introduction) Line 75: “… this initiative could foster …”
d. (Benchmarks) Line 107: “… is the one of using …”
I would suggest eliminating unnecessary phrases and redundancies to make the text clearer and more concise.

7. I have seen some sentences that, in my opinion, should be rewritten to eliminate some subjective and non-quantitative adjectives. In case any of them are necessary there must be followed by an explanation and/or a cite that support them. Some examples are:
a. (Introduction) Line 25: “… overarching …”
b. (Introduction) Line 33: “Unsurprisingly, …”
c. (Introduction) Lines 50 to 51: “… is a crucial aspect that has been mainly neglected …”.
1. Why is crucial?
2. Why has been neglected?
d. (Introduction) Line 65: “However, a worthwhile effort is the one of, …”
e. (Signature Recovery) Line 158: “Considerably less attention …”
f. (Compiler Provenance) Line 164: “… has been extensively studied.”
g. (EvalAI) Line 369: “… the well-known …”
h. (Introduction) Lines 24 to 25: “This is due to their unmatched ability to solve complex problems using a purely data driven approach.”
1. Who says this is unmatched?

8. In the Contribution, lines 81 to 82: “This design choice has been made so that researchers proficient in neural networks, but not in binary analysis can use the dataset…”. I cannot see how it is possible to validate/interpret the outputs obtained by one of these models without a basic understanding of binary analysis. I would rethink this assertion.

9. In “Data Source”, line 308 “During compilation we keep debug symbols (-g)…” I would add a comment explaining that those OBJs should not be used directly as model inputs. These binaries contain the debugging information in the DWARF-related sections and the models could use it. This is particularly important if the user of this dataset is not proficient in binary analysis. Another alternative could be to provide the stripped version of the OBJs, instead.

10. In line 70: “… e.g. understanding which compiler has generated …” I would replace “understanding” for “identifying”.

11. Lines 94 and 354. In the references to EvalAI, I would add a sentence referencing the dedicated section.

12. Line 413. I would add a description and a link to the “restricted compiler dataset” in the Massarelli et al paper: https://drive.google.com/file/d/15VUJ3iwj5VHCqAXiUcr4zJgVWSCbaU_d/view?usp=sharing

13. The text would better be reviewed by a native proofreader to eliminate some typos and minor grammar mistakes.

Experimental design

MAJOR COMMENTS

1. In the “Signature Recovery” task, to obtain the types of the function parameters, the authors propose to use a decompiler. My concern here is that the types must be collected directly from the C source code and not using a decompile. As you can see in Figure 8 of Escalada et al (2021), the accuracy of decompilers is far from perfect. So, by obtaining the type information using decompilers, the authors introduce a big source of systematic error. To get the original type of the function parameters, there are 2 options:
1. In this paper, all the programs are compiled to generate ELF binaries with debugging information. The authors can obtain the type of the parameters traversing the DWARF-related sections (https://dwarfstd.org/doc/Debugging%20using%20DWARF-2012.pdf). This is the option used by Chua et al (2017): “The function boundaries, argument counts and types are obtained by parsing the DWARF entries from the binary. Our implementation uses the pyelftools which parses the DWARF information [2]; additionally, to extract the argument counts and types, we implemented a Python module with 179 lines of code.”.
2. The other option is to traverse the source-code ASTs taking care of the includes and the typedefs. In Escalada et al (2017) you can see an example of AST traversing using Clang.

2. In the “Compiler Provenance” task, this paper mentions several publications that detect the compiler and/or the optimization level used to generate a binary using different parts of a binary:
a. Rosenblum et al (2010 and 2011) use the gaps between functions, as “these gaps may contain data such as jump tables, string constants, regular padding instructions and arbitrary bytes. While the content of those gaps sometimes depends on the functionality of the program, they sometimes denote compiler characteristics”.
b. Rahimian et al (2015) use:
* The Compiler Transformation Profile: this can be inferred from the assembly instructions.
* Strings and constants from the DATA section of the binaries.
* Literal values in the headers
* The set of additional functions included by the compiler and not present in the high-level source-code.
c. Chen et al (2019) and Massarelli et al (2019a) use only the binary code of the functions.
My concern here is that the proposed dataset does not store information that some authors have proved to be valuable to achieve this task. In my opinion, the machine learning models can improve their performance in this task if they have access to these other parts of the binary.

3. In the “Signature Recovery” task, I would like to see a discussion on the subset of types used: pointer, enum, struct, char, int, float, union and void. I suppose that the authors used the same criteria as Chua et al (2017). However, He et al (2018) use another subset: struct, union, enum, array, pointer, void, bool, char, short, int, long and long long. If any specific type is not included in the subset for any specific reason, it should be motivated.

4. Also, in the “Signature Recovery” task, I have not seen an explanation of the absence of the return type in the signature of a function. Using the example in line 294, the type of a function that sums to integers (and returns another one) is not int x int (or {int, int}) but int x int -> int (or {int, int, int}). The return type is part of the definition of the function type. In fact, in the definition of the dataset, there is a “return_type” field, but it is not used or mentioned any further in the paper. The problem of inferring the return type of a function is discussed in Escalada et al (2021).

8. I would like to suggest a task, called “Identify Function Entry Points” (or FEPs) in a stripped binary. I quote Rosemblum et al (2008) to remark on the importance of this task:
a. “The very first step in binary code analysis is to precisely locate all the function entry points (FEPs), which in turn lead to all code bytes. When full symbol or debug information is available this is a trivial step, because the FEPs are explicitly listed. However, malicious programs, commercial software, operating system distributions, and legacy codes all commonly lack symbol information.”
b. “Identifying FEPs within gaps in stripped binaries is of ultimate importance to binary code analysis …”
The following publications are related to this task:
a. Rosemblum et al (2008)
b. Bao et al (2014)
c. Shin et al (2015)
d. Escalada et al (2017) in Section 3.6
In my opinion, a model that works with binary code and is not able to identify the beginning of the FEPs will struggle to make any function-related task.


MINOR COMMENTS

1. In lines 347 to 349, it is said: “Furthermore, we have removed duplicate functions of the blind dataset. Two functions are duplicated if they contains the same sequence of instructions without considering memory offsets and immediate values.”. With this approach the models cannot be evaluated in the “Function Naming” task with functions that differ only in the literals, like:
a. bool isNewLine(char c) { return c == ‘\n’;}
b. bool isSpace(char c) { return c == ‘ ’;}
or:
c. bool isUpper(char c) { return c >= ‘A’ && c <= ‘Z’;}
d. bool isLower(char c) { return c >= ‘a’ && c <= ‘z’;}

2. In the “Function Name” metrics it is said: ”In detail, each function name is represented as a list of tokens (which are the labels to be predicted). This is obtained by splitting each function name on underscores. For example, the function name set value is splitted as [”set”, ”value”].”. The problem that I see with this approach is that there are several conventions that can be used:
a. setvalue
b. set_value
c. SETVALUE
d. SET_VALUE
e. SetValue
f. setValue
g. Even strange ones like _sEt_VaLuE_
so, the authors cannot suppose that one is the standard.

Validity of the findings

MAJOR COMMENTS

5. Regarding the usefulness and widespread adoption of this benchmark as a community standard I have 2 main comments:
a. The proposed dataset is created using only 2 compilers: GCC and Clang. The statistics (https://gs.statcounter.com/os-market-share) show that the market share of OSs that use non-ELF binaries is relevant (and even more relevant in the desktop-only market, https://gs.statcounter.com/os-market-share/desktop/worldwide). This is particularly important for the “Compiler Provenance” task. Therefore, I would like to suggest using at least the Microsoft C/C++ compiler to generate PE binary files like in Rossenblum et al (2010 and 2011) and Rahimian et al (2015).
b. For the same reason, compiling only Linux programs for an x86-64 CPU seems a bit limited. Liu et al (2018) use x86, Massarelli et al (2019b) use x86-64 and ARM binaries. To collect a big corpus of binaries that run on several OSs and CPUs, the authors might use C code generators, such as Cnerator (https://doi.org/10.1016/j.softx.2021.100711).


MINOR COMMENTS

no minor comments

Additional comments

REFERENCES

* Escalada et al (2021): https://arxiv.org/abs/2101.08116
* Cnerator: https://doi.org/10.1016/j.softx.2021.100711
* Escalada et al (2017): https://doi.org/10.1155/2017/3273891
* Katz (Deborah) et al: https://doi.org/10.1109/SANER.2018.8330222
* Katz (Omer) et al: https://doi.org/10.48550/arXiv.1905.08325
* Fu et al: https://papers.nips.cc/paper/2019/hash/093b60fd0557804c8ba0cbf1453da22f-Abstract.html
* Rosemblum et al (2008): https://dl.acm.org/doi/10.5555/1620163.1620196
* Bao et al (2014): https://www.usenix.org/conference/usenixsecurity14/technical-sessions/presentation/bao
* Shin et al (2015): https://www.usenix.org/conference/usenixsecurity15/technical-sessions/presentation/shin


I also have uploaded to the platform a PDF with the review comments sorted by importance.

Reviewer 2 ·

Basic reporting

The paper generally adheres to the basic reporting guidelines of PeerJ standards. Here are a few comments:
* Throughout the paper, I found the usage of phrases such as "representational power" of models ambiguous. I believe the authors are conflating and/or merging data representation and training models on top of the representations. Indeed, the data representation is the interface between the data and the models, but there may be various ways to evaluate the "representational power" of a model. For instance, the interpretability/explainability of an intermediate representation impacts the developer's ability to intervene in models. The authors should consider re-phrasing the associated sentences to clarify the benchmark's goals, which I believe to simply evaluate the generality of an entire deep learning model's pipeline rather than just the representation.

* The authors are encouraged to run the paper through a spell/grammar-checker as there were several typos/grammatical errors throughout the paper.

* The literature references were sufficient. However, the Related Work and Tasks sections can probably be merged. Doing so may free up more room for additional useful figures in the Tasks section. Moreover, it's unclear why the authors didn't include related work on Function Search tasks (while all the other tasks had a dedicated related work subsection). Defining each task would also help with comprehension of the related work early on.

Experimental design

The authors motivated and defined the research questions well. However, I found the metrics for the semantic tasks to be a bit too binary in terms of penalizing models, i.e., it seems there is no room for partial correctness. For instance, in the function naming task, why didn't the authors use metrics from the NLP community such as the BLEU score.

Validity of the findings

The authors did a great job identifying the impact and novelty of their work in the Discussion section--which is critical for benchmark papers. The conclusions are well stated.

Additional comments

Overall, I believe this benchmark will be very useful to the ML-binary analysis community.

---

## Round 0.2 · Minor Revisions

Reviewers are seeking minor corrections in terms of the justification provided and examples used. Please check them carefully and address the same in your revision.

·

Basic reporting

no comment

Experimental design

###
# Original review:
###

3. In the “Signature Recovery” task, I would like to see a discussion on the subset of types used: pointer, enum, struct, char, int, float, union and void. I suppose that the authors used the same criteria as Chua et al (2017). However, He et al (2018) use another subset: struct, union, enum, array, pointer, void, bool, char, short, int, long and long long. If any specific type is not included in the subset for any specific reason, it should be motivated.

###
# Authors reply:
###

R: We have used the same subset of types proposed by Chua et al (including void, that should be interpreted as ‘absence of parameters’). We have included a comment to explain it.

###
# My new review:
###

Although I understand you are using Chua et al as a reference, the aim of that paper is not to propose a standard benchmark to compare different models, as it is yours. Why the Chua et al subset is any better than any other one, like, for example, He et al? It should be motivated.

I would like to see a motivation on why any of the 14 types (bool, char, short int, int, long int, long long int, pointer, enum, struct, array, float, double, long double and void) is discarded.

For example, by using only the Chua et al types. A model with a close-to-perfect score on this task won’t be able to differentiate between:
• 2-, 4- and 8-byte integers
• 4-, 8- and 16-byte reals
However, IDA Pro which is one of the most used decompilers can separate integers and reals with different sizes. Although is true that generates a lot of FPs.
* * *
###
# Original review:
###

4. Also, in the “Signature Recovery” task, I have not seen an explanation of the absence of the return type in the signature of a function. Using the example in line 294, the type of a function that sums to integers (and returns another one) is not int x int (or {int, int}) but int x int -> int (or {int, int, int}). The return type is part of the definition of the function type. In fact, in the definition of the dataset, there is a “return_type” field, but it is not used or mentioned any further in the paper. The problem of inferring the return type of a function is discussed in Escalada et al (2021).

###
# Authors reply:
###

R: The reviewer has a point. However, the signature recovery task proposed by Chua et al (2017), which is the definition we are using, evaluates only the number of parameters and their type. Therefore, to conform with the original definition we are not using the return type.

###
# My new review:
###

The fact that Chua et al decide to exclude the return type without any further motivation does not seem enough justification to do the same. Even if it is the first paper attempting the Signature Recovery task. Both IDA Pro and Ghidra are capable of handling function return types.
* * *
###
# Original review:
###

2. In the “Function Name” metrics it is said: ”In detail, each function name is represented as a list of tokens (which are the labels to be predicted). This is obtained by splitting each function name on underscores. For example, the function name set value is splitted as [”set”, ”value”].”. The problem that I see with this approach is that there are several conventions that can be used:
a. setvalue
b. set_value
c. SETVALUE
d. SET_VALUE
e. SetValue
f. setValue
g. Even strange ones like _sEt_VaLuE_
so, the authors cannot suppose that one is the standard.

###
# Authors reply:
###

R: Thank you for the comment. We have updated the way in which we compute relevant labels for the task. We decided to use wordsegmentation (https://grantjenks.com/docs/wordsegment/ ) to break composed words in single word (as example: lt_dlloader_add -> [lt, dl, loader, add]) and it splits on camel case notation and underscore. This takes care of the cases highlighted by the reviewer. We remark that a similar approach is used in other works that tackle the function naming as this is a common problem for evaluation solutions to this problem [4].

###
# My new review:
###

Very interesting approach. In this reply, you are showing me that the segmentator is able to segment “dlloader”. However, this interesting example is not in the paper. I suggest using 3 different examples: camel case, snake case and another strange naming convention. For example, setValue, set_value and setvalue, so the reader can see all produce the same output.

Validity of the findings

###
# Original review:
###

a. The proposed dataset is created using only 2 compilers: GCC and Clang. The statistics (https://gs.statcounter.com/os-market-share) show that the market share of OSs that use non-ELF binaries is relevant (and even more relevant in the desktop-only market, https://gs.statcounter.com/os-market-share/desktop/worldwide). This is particularly important for the “Compiler Provenance” task. Therefore, I would like to suggest using at least the Microsoft C/C++ compiler to generate PE binary files like in Rossenblum et al (2010 and 2011) and Rahimian et al (2015).

###
# Authors reply:
###

R: Thank you for your comment. We have proposed two compilers (in different versions) because we focused on the Linux systems. Recall that since we are considering function level task we do not have the concept of ELF vs PE binary in our dataset. The minimal unit used in the test is the disassembled CFG of a function that does not depend on the binary format (ELF vs PE).

###
# My new review:
###

Even if your “unit of work” is the function, the generated code for any function is not the same if it is generated for Windows or for a Unix-like OS. As you can see in

https://en.wikipedia.org/wiki/X86_calling_conventions#x86-64_calling_conventions

the calling convention is not the same. It does not depend on the compiler used. If you compile the same function with CLANG/GCC (same version and compilation options) in Windows and in Linux, the resulting binary code will be different.
So, if you train a model using the binaries in the dataset (that uses System V AMD64 ABI). It will obtain much lower results evaluating functions compiled in Windows (that uses Microsoft x64 calling convention).
I suggest changing at least the title and/or the abstract to reflect the fact that the binary code of this dataset uses System V AMD64 ABI.

Additional comments

I include as an attachment a document with the original review and the new one.

---

## Round 0.3 · accepted · Accept

We appreciate your efforts in addressing the queries raised during review process. Now reviewers have recommended for acceptance of your manuscript. Congratulations.

·

Basic reporting

no comment

Experimental design

no comment

Validity of the findings

no comment

Additional comments

Congrats! It was a pleasure to review this article. I really hope this dataset to be helpful in the future.